# VFP290K: A Large-Scale Benchmark Dataset for Vision-based Fallen Person Detection

**Jaeju An**[1][*], **Jeongho Kim**[2][*], **Hanbeen Lee**[2], **Jinbeom Kim**[2], **Junhyung Kang**[2], **Minha Kim**[1], **Saebyeol Shin**[1], **Minha Kim**[2], **Donghee Hong**[1], **Simon S. Woo**[123][†]

[1]College of Computing and Informatics, Sungkyunkwan University, South Korea
[2]Department of Artificial Intelligence, Sungkyunkwan University, South Korea
[3]Department of Applied Data Science, Sungkyunkwan University, South Korea

{anjaeju, rlawjdghek, gksqls5707, kjinb1212, gogo0920, kimminha,
toquf930, sunshine01, hdh12345, swoo}@g.skku.edu

## Abstract

Detection of fallen persons due to, for example, health problems, violence, or accidents, is a critical challenge. Accordingly, detection of these anomalous events is of paramount importance for a number of applications, including but not limited to CCTV surveillance, security, and health care. Given that many detection systems rely on a comprehensive and diverse dataset, it is crucial to include fallen person images collected under diverse environments and various situations. However, existing datasets are limited to only specific environmental conditions and lack diversity. To address the challenges mentioned above and help researchers develop more robust detection systems, we create a novel, large-scale dataset for the detection of fallen persons composed of fallen person images collected in various real-world scenarios, with the support of the South Korean government. Our Vision-based Fallen Person (VFP290K) dataset consists of 294,713 frames of fallen persons extracted from 178 videos, including 131 scenes in 49 locations. We empirically demonstrate the effectiveness of the features through extensive experiments analyzing the performance shift based on object detection models. In addition, we evaluate our VFP290K dataset with properly divided versions of our dataset by measuring the performance of fallen person detecting systems. Furthermore, we discuss limitations and future work that could be extended based on our dataset. We ranked first in the first round of the anomalous behavior recognition track of AI Grand Challenge 2020, South Korea, using our VFP290K dataset, which can be found here[3]. Our achievement implies the usefulness of our dataset for research on fallen person detection, which can further extend to other applications, such as intelligent CCTV or monitoring systems. The data and more up-to-date information have been provided at our VFP290K site[4].

---

[*]Equal Contribution

[†]Corresponding author

[3]https://github.com/DASH-Lab/VFP290K

[4]https://sites.google.com/view/dash-vfp300k/

35th Conference on Neural Information Processing Systems (NeurIPS 2021) Track on Datasets and Benchmarks.

# 1 Introduction

People fall due to various causes related to indoor and outdoor accidents. In some cases, these falls can be fatal and lead to severe injuries or even death. According to World Health Organization (WHO), in the US, 20∼30% of elders who fall suffer moderate to severe injuries, such as bruises and hip fractures [3]. Moreover, there exists a golden hour to cope with, handle, and treat such falls that might cause severe injuries. If the elderly are left untreated for a prolonged time, the severity of their condition increases. Therefore, detecting fallen people in advance is crucial [1, 4, 36].

In addition, the Centers for Disease Control and Prevention (CDC)'s Child Injury Report states that about 2.8 million children visit emergency rooms across the US each year due to falls, which are the leading cause of more than 50% of baby injuries under a year old [2]. Therefore, the fallen person detector trained with our dataset can quickly notify medical staff or caregivers to handle emergencies properly and prevent further severe injuries from falls. So far, many researchers have focused on building fallen person detection models based on sensor, camera, and multimodal approaches.

Typically, sensor-based approaches are constructed based on wearable, ambient sensor, or motion capture data [16, 23, 35, 34, 28, 10], and vision-based approaches use data collected from regular or depth cameras, including CCTVs [7, 12, 22, 39, 5]. In addition, the multimodal-based approach is set up with data extracted from both sensor and camera [14, 19, 21]. Although multimodal approaches yield high and robust performance with more accurate data from people who carry multiple sensors [16], they are usually inefficient, unrealistic, and exhibit poor usability [21]. Thus, the aforementioned approaches have several practical deployment limitations, making them inappropriate for real-world applications. On the other hand, vision-based monitoring systems require minimal human intervention once they are set up. Moreover, a vision-based approach is widely used as an alternative to monitor the target area passively and further detect fallen persons. Therefore, we focus on the vision-based approach to detect fallen persons in our work.

However, existing vision-based datasets for training fallen person detection models lack the following key characteristics: light condition, camera height, and realistic occlusion effects by other objects or persons in diverse backgrounds. Primarily, existing datasets tend to be collected in particular areas such as fixed space or a small number of rooms. This results in detection models overfit to training data and perform poorly in real-world settings. In order to achieve robust and high detection performance, it requires data to be collected from diverse backgrounds, different lighting conditions.

To address these dataset challenges, we focus on constructing a novel Vision-based Fallen Person (VFP290K) dataset for detecting fallen persons, which can be used with various object detection algorithms for intelligent CCTV or monitoring applications.

Our main contributions are summarized as follows: (1) We present the VFP290K dataset, a large-scale vision-based fallen person detection dataset in real-world scenarios, providing a comprehensive public benchmark dataset for detecting fallen persons addressing the limitations of existing datasets. (2) For dataset generation, we collected 294,713 frames from 178 videos filmed at 49 different locations, representing rich backgrounds and diverse environments. In addition, we introduce and apply strict annotation rules to ensure our dataset is consistent. (3) We precisely split our dataset into training, validation, and test datasets and report statistical characteristics. (4) Finally, through extensive experiments, we empirically demonstrate the effectiveness and usefulness of our benchmark dataset, evaluating various popular object detection models.

*Note:* Our work won first place at the AI Grand Challenge 2020 (`https://www.ai-challenge.kr`) in South Korea, where the challenge was carried out to contribute to the National Dataset Generation Challenge Initiative hosted by the Ministry of Science and ICT and Institute for Information & Communications Technology Promotion, and operated by the Government of South Korea.

We publicly release our dataset[5] with a *GPL-3.0* license to foster research in this area.

---

[5]`https://sites.google.com/view/dash-vfp300k/`

Table 1: Comparison of Vision-based datasets for fallen person detection.

| Dataset | Year | Total Frames | Light condition | Camera height | # of Background | Fallen subjects in a frame | Occlusion | Availability (August, 2021) | Localization |
|---|---|---|---|---|---|---|---|---|---|
| MultiCam [7] | 2010 | 22,064 | N | 280cm | 1 (Indoor) | 1 | N | Y | N |
| Le2i [12] | 2012 | - | N | - | 4 (Indoor) | - | N | N | N |
| Mastorakis et al. [22] | 2012 | - | N | 204cm | - (Indoor) | 1 | N | N | Y |
| EDF/OCCU [39] | 2014 | 99,699 | N | - | 1 (Indoor) | 1 | Y | N | N |
| Adhikari et al. [5] | 2017 | 21,499 | N | 240cm | 5 (Indoor) | 1 | N | Y | N |
| **VFP290K (Ours)** | 2021 | 294,713 | Day: 138 Night: 40 | 1m~3m | 49 (Real-world) | 1 to 8 persons | Y | Y | Y |

## 2 Related work

Several researchers have proposed datasets for developing the fallen person detection systems [7, 12, 22, 39, 5]. These datasets are collected using various devices, such as a regular camera for an RGB image or a Kinect camera for a depth image. The summary of the existing datasets is presented in Table 1. Auvinet et al. [7] proposed the MultiCam dataset, where eight regular cameras filmed a subject in a living room. Charfi et al. [12] introduced the Le2i dataset, a collection of data acquired from a single Kinect camera in four indoor environments (home, coffee room, office, and classroom). Both datasets were proposed in the early 2010s and have drawn significant interest from researchers to develop fallen person detection.

Recently, significant advancements have been made in the fallen person detection research field. Mastorakis et al. [22] collected a depth-based dataset including 48 falls in an indoor setting taken from 204 cm height. Especially, they annotated fallen frames with a 3D bounding box for localization. Zhang et al. [39] collected two datasets, EDF and OCCU, incorporating occlusion cases for detecting fallen persons: OCCU has 80 videos containing occlusion cases, and EDF has 60 videos with only non-occlusion fall situations. Adhikari et al. [5] considered background bias by creating a fallen person dataset based on the subtracted background. On the other hand, diverse and occluded scenarios are considered only in the specific experimental setting, which cannot cover the real-world scenarios. To address such limitations, we propose the real-world 2D annotated vision-based dataset, VFP290K, incorporating diverse background, light, location, and scene conditions as well as different human activities.

There are numerous studies for detecting fallen persons using computer vision techniques [32, 29, 11, 38, 17]. While early approaches used heuristic-based methods that compare the ratio between the width ($W$) and height ($H$) of the detected bounding box, recent works utilize deep learning, such as pose estimation or distance measurement with 3D depth images [6]. However, so far, comparing the performance of different approaches has not been easy due to the lack of benchmark datasets. To address this issue, we propose a fallen person dataset as a standard benchmark that can be evaluated across different algorithms. Also, our dataset provides 2D annotated information along with the expected benchmark performance from popular detection algorithms.

## 3 Vision-based Fallen Person (VFP290K) Dataset

### 3.1 Overview

We introduce VFP290K dataset, the novel benchmark dataset for the fallen person detection. Our VFP290K is comprised of 294,713 frames extracted from 178 videos, including 131 scenes in 49 locations, capturing several essential aspects needed for a detection task such as different light condition and camera height. Table 1 describes the details of our dataset compared to existing datasets. Specifically, we define metadata as follows, where the more detailed explanation can be found in Appendix 1.

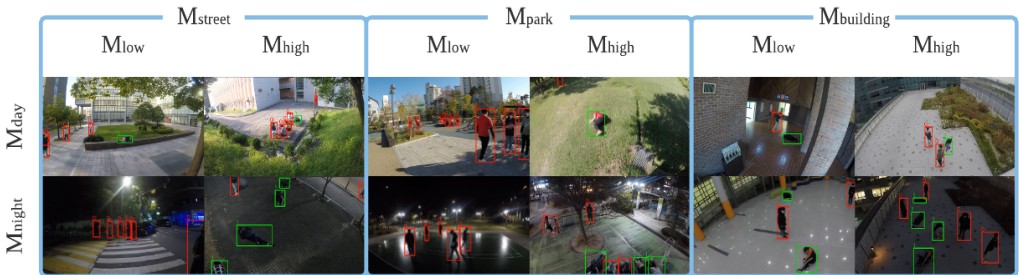

Figure 1: Samples of the annotated images in the VFP290K dataset. A red bounding box indicates a normal person, and a green bounding box indicates a fallen person.

**Light condition** ($M_{day,night}$)**.** Light condition is one of the most important features impacting the fallen person detection performance, as the quality of the image is highly dependent on it. We consider two different light conditions as follows: $M_{day}$ and $M_{night}$.

**Camera height** ($M_{low,high}$)**.** Camera height is directly related to CCTV and indoor camera environments. It is consequential to model contemplate all the possible height requirements for different CCTV devices, as there are many different types of CCTV with varying positions. To address this issue, we film the video into two different camera heights with $M_{low}$ and $M_{high}$, representing about $1 \sim 3$m and higher than 3m, respectively.

**Background** ($M_{street,park,building}$)**.** One of the main contributions of our work is to include a number of different backgrounds, from public areas to indoor environments. We divide the background into the following three sub-categories: street, park, and building, whereas other datasets capture the fixed or same location.

**Location** ($M_{location}$)**.** Along with the three background categories, we vary and specify each filming region to different locations, where the videos are shot in 49 locations. Therefore, we can provide more fine-grained and detailed information about the background features.

**Scene** ($M_{scene}$)**.** As a subcategory of each location, we enumerate entire scenes divided by different viewpoints in whole 178 videos. That is, some videos have multiple viewpoints in the exact location.

### 3.2   Dataset Generation

We generate our VFP290K dataset with the following three steps: data acquisition, data annotation, and data quality assurance.

**Data acquisition.** We use GoPro HERO5 camera, which provides diverse resolution settings similar to CCTV with a 1080p wide-angle lens. The image size is configured to $1920 \times 1080$. We collected data for two months, from August to October 2020. And we recruit 15 volunteers to perform different fall scenarios, where the recruitment and consent form details are provided in Appendix 2. In order to provide a more comprehensive dataset, we attempted our best to include more diverse ethnicity. In particular, three students are international students with different skin colors and age. Also, to provide a more diverse dataset, volunteers intentionally change their appearance by changing clothes, wearing masks and caps. Given a set of scenarios at different locations, participants performed walking, roaming, and falling in streets, parks, and indoor areas. Also, we perform face anonymization for pedestrians who are not directly participated in our study. The detailed explanation for the anonymization can be found in Appendix 5, and the samples of our dataset are presented in Fig. 1.

**Data annotation.** First, we consider the object of our dataset to be either a fallen and non-fallen person. We assign "class 1" to a bounding box of a fallen person and "class 0" to that of a non-fallen person, constructing our fall person object detection task as a binary classification task. Precisely, we define the fall events considering the different postures followed by stumbling, bending down, and limping, similarly shown in the work proposed by Foroughi et al. [15].

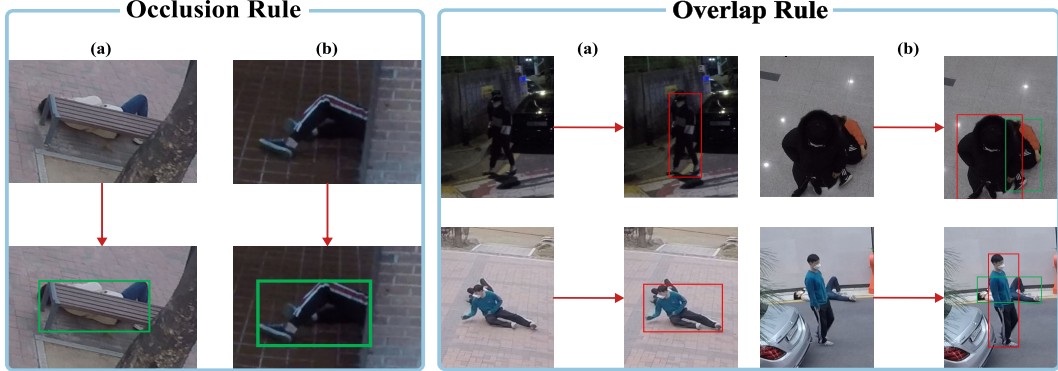

Figure 2: The rigorous annotation rules for the high-quality and consistent dataset. Occlusion Rule (a): We assign a bounding box to the entire body even if a target person is partially occluded. Occlusion Rule (b): Since an object occludes the upper body, we only annotate the detectable body parts: legs. Overlap Rule (a): We do not annotate the person at the back since the front person's frame occludes the other subject's frame over 80%. Overlap Rule (b): both persons are annotated because the occluded part is less than 80%.

Moreover, we use the following three rules to determine the fall event: 1) both shoulders should touch the ground (representing lie down, backward, and forward fall), 2) one side shoulder touches the ground (representing sideway fall), and 3) the lower body from hips to foot touches the ground when the upper body is occluded (sitting down from fall). Next, we annotate the bounding box to localize each subject using the LabelImg annotator [31]

In order to provide accurate annotations as much as possible, including overlap and occlusion cases, we introduce strict annotation rules, which ensure the consistency and high-quality of the VFP290K dataset. The details of the overlap and occlusion cases can be found in Appendix 4. Two strict annotation rules are enforced to provide accurate and consistent labeling as follows:

- **Rule 1. Occlusion labeling rule.** We assign a bounding box to the entire body of a target person when other objects or persons partially occlude the target person's body (referring to the occlusion rule (a) in Fig. 2). In addition, we annotate the bounding box only at the visible body parts of the subject if the subject of the frame is severely occluded so that the other part of the body is invisible (referring to the occlusion rule (b) in Fig. 2).

- **Rule 2. Overlap labeling rule.** Overlap is a special case of the occlusion case, where the object is blocked by a person, inducing two bounding boxes to cross over. In this case, we only allocate a bounding box to the person in the front when the person in the back is occluded more than 80% (See the overlap rule (a) in Fig. 2). Otherwise, we annotate all persons based on the occlusion rule (see overlap rule (b) in Fig. 2). Note that we only apply the overlap rule to the not fallen cases, as our VFP290K dataset focuses on fallen person detection.

**Data quality assurance.** We assure the high-quality of the VFP290K dataset based on a rigorous quality control process. We employed five different students from our institution to cross-check every single frame for each video, where each student was instructed to adhere to the following annotation check rules to determine if: 1) in the given frame, the occlusion labeling rule and overlap labeling rule are strictly enforced, 2) a bounding box is correctly shown for the fallen person according to the fall event definition, 3) the label for the fall and non-fallen frames are correctly assigned, and 4) personally identifiable information (PII) is fully anonymized for non-volunteers. We provide separate training for each student to make sure they understand the specific rules, by first explaining the purpose of the dataset and highlighting the details of the annotation process over several days. If there is a disagreement, all five annotators discussed and reached an agreement when at least half of them agreed.

### 3.3 Dataset Analysis

**Num. of different metadata features.** We provide the number of videos, frames, places per background, and scenes per background, as shown in Table 2, to capture the underlying distribution of each metadata feature. Note that $M_{scene}$ is calculated by counting the combination of different places and views. We maintain the balance of two types of camera height that are under or over 3m. We mainly install the camera on the street rather than installing it in a park or building since the CCTV is likely to be more popularly used in the street environment. In addition, we change viewpoints, producing 29, 95, and 7 scenes, respectively, to provide more diverse background information.

Table 2: The number of data according to features in our VFP290K dataset.

| Features | | # of videos | # of frames | # of places | # of scenes |
|---|---|---|---|---|---|
| **Light Condition** | Day | 138 | 221,666 | - | - |
| | Night | 40 | 73,047 | - | - |
| **Camera height** | Low | 97 | 50,081 | - | - |
| | High | 81 | 244,632 | - | - |
| **Background** | Street | 118 | 57,965 | 13 | 29 |
| | Park | 47 | 212,752 | 30 | 95 |
| | Building | 13 | 23,996 | 6 | 7 |

**Analysis on feature-level distribution.** We assume that if the dataset possesses diverse underlying characteristics about background or light condition, then the dataset would be much more distributed than clustered. Thus, we analyze the feature level distribution of our dataset by visualizing and validating the underlying distribution of high-dimensional metadata features. Furthermore, we compare the feature distributions between ours and other popular publicly available datasets (The more detailed explanation can be found in Appendix 6). As shown in Fig. 3.(a) and (b) for MultiCam and Adhikari et al.'s dataset showed the clustered data points at each region, implying their dataset is limited in diversity. On the other hand, our VFP290K dataset are more distributed across different data points, as shown in Fig. 3.(c) and (d). These clearly visualize and demonstrate that our dataset is more suitable for developing a robust detection system, as our dataset captures more diverse condition than other datasets.

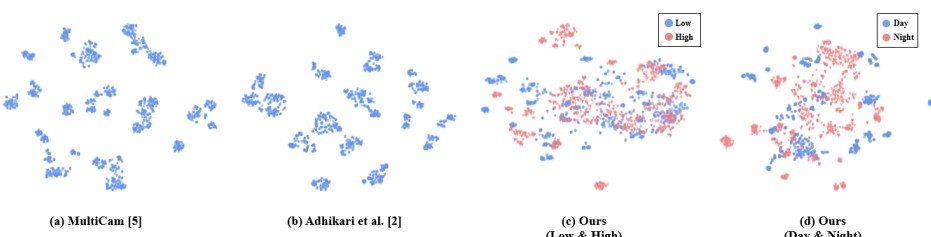

Figure 3: Visualization of the datasets from feature perspective. We utilize ImageNet-pretrained VGG-19 network [27] to obtain feature vectors and t-SNE [33] results. Each data point indicates an image sample.

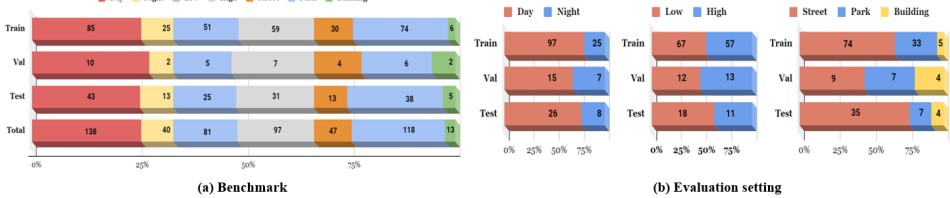

Figure 4: The distributions of our VFP290K benchmark and evaluation setting.

### 3.4 Benchmark Dataset

Creating a quantifiable and comparable benchmark dataset is one of our main contributions by providing VFP290K dataset composing training, validation, and test dataset. An unbiased and non-redundant split in a benchmark dataset is a crucial issue. Therefore, we proceeded with a balanced video-wise split, where the total number of frames/videos with different

characteristics and scenarios is roughly maintained in the same ratio to prevent bias. Moreover, the training and test split is maintained in 8:2 ratio. Indeed, we presented our balanced video-level statistics in Fig. 4.

The summary of the benchmark dataset information is presented in Fig. 4.(a). Training and test datasets are divided into the ratio of 8:2 based on the full frames, covering all features, such as light condition, camera height, and background. More specifically, training, validation, and test datasets have a total of 216,104, 19,085, and 59,525 frames, respectively. In addition, we also consider the statistically balanced data split for our evaluation setting for each set. As shown in Fig. 4.(b), all separated datasets have a balanced distribution split in the same way for the benchmark. Note that we provide 4 JSON files representing each data split information for reproducibility. As a result, all datasets, including training, validation, and test, have a balanced distribution across all features in our VFP290K dataset.

### 3.5 Dataset Availability, Usage, and Ethics

**Dataset availability.** We manage and provide our VFP290K dataset through our VFP290K website[6], where VFP290K dataset is maintained and available on our website's download tab[7]. Also, we release our dataset under a *CC BY 4.0 license.*

**Dataset usage.** Our data can be utilized for a variety of purposes in the area of a fallen person or anomalous event detection and further be used as a part of the crime/disaster prevention system. For example, intelligent CCTV systems have become an essential for social infrastructures installed in streets, parks, and buildings for security and health care applications. More than 1 billion CCTV cameras are expected to be installed worldwide [30] to capture and provide rich information about human activities, such as to capture the evidence of crimes and monitor health incidents. However, manually identifying anomalous events in the recorded videos is a highly time-consuming task. Therefore, it is critical to leverage machine learning algorithms to automatically detect such anomalous events. In addition to the benchmark dataset, we provide the several fallen person detectors and release our code here[8] with a *GPL-3.0* license.

**Ethics and social impact.** Our VFP290K dataset adheres to the standards of FAIR Data principles [37] to meet findability, accessibility, interoperability, and reusability so that all relevant metadata is specified to the research community. For volunteers in our dataset, we explained the purpose of our data collection and received the agreement from individuals in advance. Also, we completely anonymized the PII of non-volunteers using the anonymization methods described in Appendix 5. Furthermore, our work does not have a negative social impact, as we fully adhered to the data collection protocol approved by the IRB and anonymized personal information.

## 4 Experiments

### 4.1 Experimental Setting

We conduct two different experiments with our VFP290K dataset and illustrate how performance changes according to various characteristics. The first experiment evaluates the overall performance of the benchmark on VFP290K dataset using different baseline models. For the baseline models, we use the following 7 popular object detection models: Two-stage detection models (Faster-RCNN [26], Cascade R-CNN [8], and DetectoRS [24]), one-stage detection models (RetinaNet [20], YOLOv3 [25] and YOLOv5 [18]), and transformer-based model (DETR [9]), where more detailed model descriptions are provided in Appendix 7. And, the details of training, validation, and test distribution are shown in Fig. 4.

The second experiment evaluates the detection performance with different environment conditions, as we hypothesize that certain environment (e.g., night) would be more challenging to detect fall person than other cases (e.g., day). Specifically, we consider the following

---

[6]`https://sites.google.com/view/dash-vfp300k`
[7]`https://doi.org/10.23056/VFP300K_DASHLAB`
[8]`https://github.com/DASH-Lab/SwoonDetector`

Table 3: Benchmark performance of VFP290K dataset. We conduct experiments in 7 popular object detection models: Two-Stage (Faster R-CNN (F R-CNN), Cascade R-CNN(C R-CNN), DetectoRS), One-Stage (RetinaNet, YOLO3, YOLO5) and DETR in Transformer-Based (T-Based).

| Method | Two-Stage | | | One-Stage | | | T-Based |
|---|---|---|---|---|---|---|---|
| Model | F R-CNN | C R-CNN | DetectoRS | RetinaNet | YOLO3 | YOLO5 | DETR |
| $mAP$ | 73.2 | 75.1 | 74.6 | 75.0 | 59.0 | 74.1 | 60.5 |
| $AP_{50}$ | 87.3 | 87.4 | 86.6 | 91.0 | 81.3 | 83.8 | 86.8 |
| $AP_{75}$ | 79.9 | 81.1 | 79.7 | 81.1 | 67.0 | 78.4 | 68.7 |

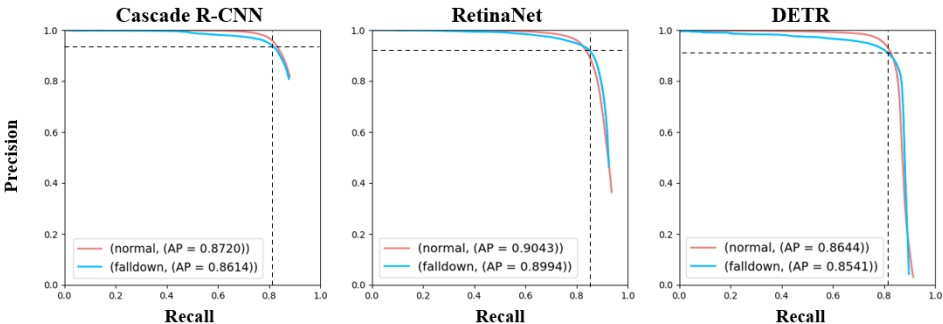

Figure 5: The Precision-Recall curves for Cascade R-CNN, RetinaNet, and DETR on the benchmark dataset. $AP$ represents $AP_{50}$ score, and the dotted line indicates the point where $AP$ drops drastically for fallen person class.

scenarios: $M_{day}$ vs. $M_{night}$, $M_{low}$ vs. $M_{high}$, and $M_{street}$ vs. $M_{park}$ vs. $M_{building}$. As mentioned previously, the purpose of this experiment is to identify the effect of various feature characteristics. We sample one frame per every five frames for each experiment for training, as this experiment is to demonstrate the effect of the specific features, which cannot be captured in the first baseline experiments. For the detail of training, validation, and test distribution, they are presented in Table 2 and Fig. 4.(b). Note that we conduct the second experiment mainly for one-stage detection models, as they can be easily utilized for the real world deployment.

**Evaluation metrics.** We use the average precision ($AP$) over multiple Intersection over Union ($IoU$) thresholds as our evaluation metric. Specifically, we use averaged $mAP$ with $IoU$ thresholds, ranging from 0.5 to 0.95 with a step size of 0.05. $AP$ values are calculated with respect to a single $IoU$ threshold (0.5, 0.75), denoted as $AP_{50}$ and $AP_{75}$. Also, we report precision-recall curves (PR curves) to capture the trade-off between the precision and recall for the benchmark performance.

**Implementation details.** To ensure reproducibility and maximize our dataset accessibility, we leverage as much publicly accessible implementations as possible. We choose the official implementations based on MMDetection toolbox [13] for the baseline models except YOLOv5. As YOLOv5 is not supported yet, we use the official implementation from here[9]. The more detailed hyper-parameter settings and configurations for the first and the second experiment are provided in Appendix 7.

## 4.2 Experiment 1. Benchmark Performance

The performance results are presented in Table 3. While two-stage models such as Faster R-CNN, Cascade R-CNN, and DetectoRS achieved $mAP$ scores of 73.2, 75.1, and 74.6, respectively, one-stage models such as RetinaNet, YOLOv3, and v5 produced $mAP$ scores of 75.0, 59.0, and 74.1. Furthermore, DETR achieved 60.5 $mAP$ which is drastically decreased from 86.8 $AP_{50}$. The best performing model is Cascade R-CNN based on $mAP$. As a result,

---

[9]https://github.com/ultralytics/yolov5

Table 4: Experimental results for comparing performance change according to different background conditions. We present performance by altering the training and test set for $M_{street}$ vs. $M_{park}$ vs. $M_{building}$ scenario. These experimental results demonstrate that diverse backgrounds make a robust detection model for fallen events. Bold text indicates the highest performance.

| Backbone | Training Test | Street | Park Street | Building | Street | Park Park | Building | Street | Park Building | Building |
|---|---|---|---|---|---|---|---|---|---|---|
| Faster R-CNN | $AP$ | 74.2 | 73.2 | 61.6 | 62.0 | 70.6 | 51.7 | 74.8 | 84.7 | 70.2 |
| | $AP_{50}$ | 91.0 | 86.0 | 82.8 | 78.6 | 85.7 | 70.5 | 87.6 | 95.7 | 82.1 |
| | $AP_{75}$ | 82.9 | 80.9 | 72.3 | 69.0 | 76.8 | 58.8 | 81.3 | **92.0** | 79.1 |
| RetinaNet | $AP$ | **77.0** | 74.3 | 65.4 | 66.4 | **73.7** | 58.7 | 82.8 | **85.1** | 80.4 |
| | $AP_{50}$ | **92.2** | 86.1 | 81.1 | 83.0 | **88.8** | 75.2 | 93.2 | **96.0** | 91.5 |
| | $AP_{75}$ | **84.3** | 80.4 | 73.0 | 72.0 | **79.1** | 64.7 | 90.1 | 91.8 | 87.5 |
| YOLOv3 | $AP$ | 61.0 | 51.0 | 28.4 | 41.6 | 53.7 | 28.2 | 61.0 | 66.4 | 67.1 |
| | $AP_{50}$ | 81.7 | 66.4 | 40.0 | 57.8 | 75.9 | 42.1 | 81.7 | 82.4 | 83.1 |
| | $AP_{75}$ | 68.9 | 60.0 | 33.6 | 46.8 | 63.2 | 31.5 | 68.9 | 78.4 | 79.0 |
| YOLOv5 | $AP$ | 66.9 | 67.1 | 22.6 | 39.8 | 69.2 | 20.9 | 67.5 | 80.2 | 60.6 |
| | $AP_{50}$ | 78.3 | 74.5 | 33.5 | 46.5 | 77.6 | 33.5 | 74.3 | 84.8 | 70.7 |
| | $AP_{75}$ | 72.9 | 71.9 | 26.6 | 42.8 | 72.7 | 26.6 | 72.7 | 83.6 | 67.9 |

Table 5: Experimental results for comparing performance change according to light condition and camera height. We present the performance by altering the training and test set for $M_{day}$ vs. $M_{night}$ or $M_{low}$ vs. $M_{high}$ scenario. This experimental result demonstrates that balancing light condition and camera angle images is important to make a robust detection model for fallen events. Bold text indicates the highest performance.

| Backbone | Training Test | Day | Night Day | Day | Night Night | Low | High Low | Low | High High |
|---|---|---|---|---|---|---|---|---|---|
| Faster R-CNN | $AP$ | 76.7 | 63.2 | 52.3 | 55.9 | 70.0 | 57.3 | 56.1 | 72.9 |
| | $AP_{50}$ | 91.7 | 82.6 | 71.4 | 78.3 | 89.8 | 76.0 | 74.9 | 89.6 |
| | $AP_{75}$ | 84.3 | 80.8 | 57.2 | 60.9 | **80.8** | 66.9 | 63.6 | **81.7** |
| RetinaNet | $AP$ | 77.9 | 66.7 | 53.4 | **56.6** | 70.2 | 61.0 | 59.6 | **73.9** |
| | $AP_{50}$ | **93.2** | 85.6 | 74.7 | **78.5** | **90.3** | 81.8 | 78.0 | **90.9** |
| | $AP_{75}$ | **84.8** | 74.1 | 56.7 | **62.0** | **79.2** | 69.5 | 66.9 | **81.7** |
| YOLOv3 | $AP$ | 61.5 | 43.2 | 29.9 | 41.5 | 56.7 | 37.5 | 34.9 | 56.3 |
| | $AP_{50}$ | 87.4 | 63.0 | 54.5 | 63.5 | 80.8 | 60.6 | 53.0 | 80.0 |
| | $AP_{75}$ | 72.8 | 49.0 | 30.6 | 45.1 | 67.8 | 41.4 | 39.4 | 65.3 |
| YOLOv5 | $AP$ | **79.4** | 34.3 | 39.2 | 41.4 | 59.0 | 41.2 | 35.0 | 71.8 |
| | $AP_{50}$ | 88.8 | 44.7 | 51.7 | 56.1 | 75.2 | 54.2 | 44.8 | 84.3 |
| | $AP_{75}$ | 84.2 | 38.4 | 41.6 | 44.2 | 68.0 | 46.5 | 39.4 | 78.1 |

two-stage models performed better than one-stage models or DETR; however, one-stage models except YOLOv3 still achieved comparable performance to other models.

We present the PR curves for the best performing models in each method: Cascade R-CNN, RetinaNet, and DETR. As shown in Fig. 5, each PR curve is created by ranking the detection results exploiting the classifier's scores and thresholding values. When recall approaches 0.83, precision drops drastically, indicating that the models are confused to detect objects based on high recall scores. This can be a critical issue for the fallen person detection, as the ultimate goal is to detect people who have fallen. Thus, we show that it is important to develop a detection model that focuses on improving recall. Overall, to the best of our knowledge, there are no previous datasets annotated with a bounding box for vision-based fallen person detection. And, we provide the standard benchmark dataset performance using VFP290K dataset with the popular detection models.

## 4.3 Experiment 2. Performance under Different Environment Conditions

**Different background.** We hypothesize that the performance of the fallen person detection system can be affected by background bias. Therefore, we perform a series of experiments that measure performance by altering the $M_{street}$, $M_{park}$, and $M_{building}$ scenarios. The performance is compared in Table 4. When evaluating the models trained with each background, they performed well on the identical type of background, except the $M_{building}$. Also, we observe the similar trends that the models showed significant performance drop,

when deploying other types of background, e.g., Faster R-CNN trained with $M_{street}$ produced 74.2 $mAP$ on $M_{street}$'s test set, but obtained 62.0 $mAP$ on $M_{park}$'s test set.

Interestingly, regardless of the dataset the models was originally trained, $M_{building}$ is relatively easy to detect due to fewer people appearance. Since VFP290K dataset has various distributions for background, models trained on our VFP290K dataset can be more robust against background bias due to more diverse environments we provide.

**Different light condition and camera height.** We conduct experiments on different light condition ($M_{day}$ vs. $M_{night}$) and camera height ($M_{low}$ vs. $M_{high}$). The performance results are presented in Table 5. Interestingly, the models trained with $M_{night}$ produced similar performances on both light conditions. This result implies the model trained with $M_{night}$ can produce more robust and resilient results with varying light condition and camera height. Interestingly, at night, we learned that people's skin color is less critical for detection models because of the dark luminescence. On the other hand, the silhouette of human posture is more important to detect the fall event.

## 5    Discussion

To the best of our knowledge, we are the first to create a large-scale vision-based fallen person dataset covering diverse environments, representing multiple view, different occlusion and light condition with simulating realistic situations. Although our VFP290K dataset shows great promise for research purposes, we encountered several limitations while creating and experimenting with the dataset.

**Limitations.** First, despite the broad coverage of various features under realistic situations, the VFP290K dataset did not consider the falling scenarios from physical violence. Second, we did not focus on too tiny objects, due to the difficulty of annotation. Also, we did not consider all different weather environments, such as rainy or storm weather condition. Lastly, we could not include all the age groups, but our work includes diverse backgrounds and ethnicity. As our dataset will be continuously updated, we plan to include different age groups for future work.

**Anomalous Event Detection.** We conducted an additional experiment using the best performers of our VFP290K, Cascade R-CNN, and RetinaNet, to detect an anomalous event defined as a fallen situation. As a result, we confirmed that the performance of anomalous detection is slightly worse than training both fallen and non-fallen persons. The experiment showed that our dataset could also be used for anomalous event detection. The details are provided in Appendix 8.

**Future work.** We plan to develop an extended version of the VFP290K dataset comprising images collected from abnormal user behavior scenarios and collect more data, considering more diverse weather condition and including tiny objects. In addition, we consider deploying the detection models for real publicly available fall test videos (from YouTube) to verify the generalizability of our dataset.

## 6    Conclusion

We propose a large-scale vision-based fallen person dataset, VFP290K, consisting of 294,713 frames extracted from 178 videos. To the best of our knowledge, our data is the largest dataset, incorporating the diverse conditions. Also, we applied strict and consistent annotation rules to produce a high-quality dataset. Furthermore, we present benchmark results for vision-based fallen person detection using the popular object detection algorithms. We demonstrate that detecting fallen persons from the VFP290K dataset is challenging, as it successfully reflected real-world conditions, and we further find that the performance can vary according to differing environments. Moreover, we present a side-by-side comparison between VFP290K and existing dataset, showing much more rich and diverse features through extensive analysis and evaluation. Overall, our VFP290K dataset shows great promise for real-world applications related to fallen person detection using CCTV or monitoring systems.

## Acknowledgments and Disclosure of Funding

This work was partly supported by Institute of Information & communications Technology Planning & Evaluation (IITP) grant funded by the Korea government (MSIT) (No.2019-0-00421, AI Graduate School Support Program (Sungkyunkwan University)), (No. 2019-0-01343, Regional strategic industry convergence security core talent training business) and the Basic Science Research Program through National Research Foundation of Korea (NRF) grant funded by Korea government MSIT (No. 2020R1C1C1006004). Also, this research was partly supported by IITP grant funded by the Korea government MSIT (No. 2021-0-00017, Original Technology Development of Artificial Intelligence Industry), and was partly supported by the MSIT (Ministry of Science, ICT), Korea, under the High-Potential Individuals Global Training Program (2020-0-01550) supervised by the IITP (Institute for Information & Communications Technology Planning & Evaluation).

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
