# OpenReview forum: "VFP290K: A Large-Scale Benchmark Dataset for Vision-based Fallen Person Detection"
_NeurIPS.cc/2021/Track/Datasets_and_Benchmarks/Round2 — NeurIPS 2021 Datasets and Benchmarks Track (Round 2)_

### Official Review · Reviewer_MQ3T · 2021-09-18
**The paper compiles a fallen-person benchmark in various settings**

**Rating:** 7
**Confidence:** 4
**Correctness:** Seems to be correct.

**Strengths:**

The dataset seems to be the largest and the most varied from the literature. The experimental evaluation shows that on this dataset the problem of detecting fallen persons can be solved with moderate success by state-of-the-art object detectors (Faster-RCNN, RetinaNet, YOLO). The experimental evaluation with this respect seems to be well carried.

**Weaknesses:**

-My main concern in building this dataset is that it contains videos "performed" artificially by the volunteers. It is not clear that a model trained on this dataset will generalize well on real cases, not performed.

-It would have been nice to see some experiments also on the anomalous event detection task.

**Additional Feedback:**

See weaknesses.

**Clarity:**

The paper is written mostly clear, there are some sentences that sound a little bit awkward:
-"Given that many detection systems rely on a comprehensive dataset comprising fallen person images collected under diverse environments and in various situations is crucial."
-"We weigh on the street than a park or building for the background since the CCTV is likely to be installed more on the street environment."

**Documentation:**

Seems to be well documented.

**Ethics:**

N/A.

**Relation To Prior Work:**

Seem to be well addressed.

**Summary And Contributions:**

This paper introduces a large-scale dataset Vision-based Fallen Person (VFP290K) for the detection of fallen persons, It consists of
294,714 frames of fallen persons extracted from 178 videos, including scenes in 49 locations. This is 3x more than [35]. Also the dataset does vary in light condition (day vs night), camera height (low vs high)and  background (street vs park vs building). The authors investigate in the experimental section the usefulness of their dataset for the task of binary classification between fallen and not-fallen person in several experimental settings.

---

> ### Author Response · Authors · 2021-09-27
> **Response to Reviewer MQ3T (Part2)**
>
> We thank the reviewer for the great comment and insightful suggestion. Indeed, our approach and dataset can be directly applicable and be extended for the video anomaly detection task, where the anomaly is the fall event in our case. Since we have a label specifically for the fall (anomalous event), we can evaluate the latest video anomaly detection algorithms for benchmarking. This would be a great extension, and we will try our best to make our benchmark dataset available for the video anomaly detection task. In the final version, we plan to include the evaluation of video anomaly detection algorithms.
>
> In addition, we fixed the sentences that required clarifications by the reviewer as follows in the revised version of the paper:
>
> "Given that many detection systems rely on a comprehensive and diverse dataset, it is crucial to include fallen person images collected under diverse environments and in various situations."
>
> "We install the camera on the street rather than a park or building, since the CCTV is likely to be more popularly used in the street environment."
>
> Once again, we sincerely thank the reviewer for their helpful comments, suggestions, and valuable feedback.

---

> > ### Comment · Reviewer_MQ3T · 2021-09-30
> > **Rating update**
> >
> > Based on the author's responses addressed to me and to the other reviewers I have increased by score to 7. Good luck!

---

> ### Author Response · Authors · 2021-09-27
> **Response to Reviewer MQ3T (Part1)**
>
> We appreciate the constructive comment. We fully understand that the fallen acts from the volunteers would be slightly different from the actual fallen events. However, we attempted our best to emulate the real and natural fallen events from volunteers over realistic and diverse environments, locations, lighting conditions, backgrounds, and ground conditions, in varying degrees of scenarios. As shown in the video samples on our website (https://sites.google.com/view/dash-vfp300k/), we believe that there is no significant difference between our sample videos and real-life situation.
>
> Overall, we believe, though ours is staged, our dataset provides the essential characteristics of the fall events and even includes more challenging fallen scenarios, including occlusion, different fall postures (before/after fall), which are essential to training the OD model for fallen person detection. These are not addressed in the prior benchmark dataset.
>
> In addition, the generalizability of our datasets and algorithms are demonstrated against the real fall test dataset during AI Grand Challenge sponsored by the Korean government, winning 1st place. In the final version, we plan to include the result of our approach by testing real publicly available fall test videos (from YouTube, etc.) to demonstrate the generalizability, practicality, and usefulness of our dataset.

---

### Official Review · Reviewer_qsU8 · 2021-09-19
**Great dataset for more robust fallen person detection, but limited broader impact**

**Rating:** 5
**Confidence:** 3
**Clarity:** Yes

**Strengths:**

- the dataset serves a clear purpose and facilitates study of an important application of object detection systems
- the authors conduct a comprehensive study of baseline object detectors and publicly release their data, the evaluation code and the trained baseline models
- the dataset seems to be substantially more diverse than existing datasets in this area

**Weaknesses:**

- the challenges that this dataset addresses are important for the specific application of fallen person detection, however they don't seem to facilitate more fundamental study of the challenges that arise in the underlying algorithms that are used here (namely deep learning based object detectors). The algorithmic challenges that are of interest to the Neurips community (limited generalization due to lack of background / lighting / viewpoint / occlusion diversity in the training data) have been addressed by many large-scale object detection benchmarks in the past.
- While the background and lighting conditions are diverse, the set of persons that perform the fall scenarios seems quite small (15). Here, the algorithms could still be prone to varying appearance across humans (e.g. elderly vs young, different skin color).
- While i understand that it can be challenging to obtain real falling person scenes, and it is much more efficient to let volunteers perform the fall scenarios, it has the disadvantage that an algorithm performing well on exclusively acted fall scenarios might not generalize to real footage of falling persons. Hence it would be good to rely on real footage at least for the test data.

**Additional Feedback:**

The dataset might be better suited for an application oriented computer vision audience.

**Correctness:**

Yes, except for the two points:
- only 15 volunteers do not capture the varying appearance of humans
- all fall scenarios are staged, which might look different from real footage of fallen persons

**Documentation:**

Yes, the authors provide a lot of details on the data collection process and the dataset is easily accessible as well as the accompanying models / evaluation code. It would be great to have a public leaderboard on the website though.

**Ethics:**

No, sensitive personal information like license plates and faces have been anonymized.

**Relation To Prior Work:**

Yes

**Summary And Contributions:**

This paper introduced a large-scale dataset for studying the problem of fallen person detection. Unlike existing datasets, the authors collect data in diverse environmental and lighting conditions. They collect their data with a single GoPRO HERO 5 camera and 15 volunteers that perform different fall scenarios. The data was collected in 39 different locations in South Korea, totaling 178 video clips. This way, the authors aim to address the existing limitations of existing methods for falling person detection which lack robustness to environmental and lighting conditions. The dataset contains annotations for both fallen and non-fallen persons.

---

> ### Author Response · Authors · 2021-09-27
> **Response to Reviewer qsU8 (Part3)**
>
> We thank the reviewer for the constructive comment. First of all, we fully understand that the acted fallen events may be different from the actual fallen events; however, we tried our best to emulate the real and natural fallen events from volunteers, as shown in the sample videos (https://sites.google.com/view/dash-vfp300k/). As shown in the samples, we believe there are no enormous differences between our sample videos and real-life falls.
>
> Indeed, the generalizability of our datasets and algorithms are demonstrated against the real fall test dataset during the AI Grand Challenge sponsored by the Korean government, where we won 1st place. In the final version, we plan to include the result of our approach by testing real publicly available fall situation videos to demonstrate the generalizability, practicality, and usefulness of our dataset.

---

> ### Author Response · Authors · 2021-09-27
> **Response to Reviewer qsU8 (Part2)**
>
> In order to provide a more comprehensive dataset, we attempted our best to include more diverse ethnicities. In particular, three students are international students with different skin colors. In order to provide a more diverse dataset, volunteers intentionally change their appearance by changing clothes, wearing masks, and caps, etc. Also, non-volunteers who are in the background captured varying ages of people. We added this in Section 3.2.
>
> Interestingly, at night, we learned that people's skin color and age are less critical for object detection (OD) algorithms because of the dark luminescence. On the other hand, we found out that the silhouette of human posture is more important for the OD algorithms to detect the fall event. Once again, we acknowledge that we did not include all the age groups in our dataset, but our work attempted to include diverse backgrounds and ethnicity. Our dataset will be continuously updated. Hence, we plan to include different age groups for future work. We will discuss this limitation in Section 5.

---

> ### Author Response · Authors · 2021-09-27
> **Response to Reviewer qsU8 (Part1)**
>
> We thank the reviewer for the question. From the careful literature reviews and prior similar dataset investigation, we have learned that there is a significant gap in providing more practical and diverse datasets. Moreover, most of the existing datasets lack diversity in the environment, the number of people, background, etc.
>
> We clearly demonstrate their limitation in Fig. 3. The number of fallen actors (up to 15) in a frame sounds small, but ours is by far the largest compared to the prior dataset. Also, there are additional people in the background (anonymized non-volunteers) naturally walking or passing by. Therefore, we believe that we have covered the diversity as well as natural activities from non-volunteers, providing the foundation to train better and robust object detection neural network algorithms.

---

### Official Review · Reviewer_1eW1 · 2021-09-19
**A large dataset with high variety for fallen person detection.**

**Rating:** 7
**Confidence:** 4

**Strengths:**

-	The dataset achieves 1st place at the AI Grand Challenge 2020 in South Korea, which is operated by the Government of South Korea.
-	The dataset is the largest up-to-date for fallen person detection, significant larger than the 2nd largest dataset (2x).
-	The dataset considers various conditions (day/night, camera heights, backgrounds) that other datasets lack.


**Weaknesses:**

-	Although demonstrate clear rules (Occlusion Rule and Overlap Rule) for quality control for the annotation procedure, there still exists ambiguities:
o	How would annotator categorize people during their falls (not completely on the ground)? There should be a clear definition for this binary classification problem.
o	Authors said the annotations for each frame are cross-check between 5 different students but did not specify the rule for interrater agreement and the criteria to accept an annotated frame.
-	As the dataset is composed of frames from multiple videos, there might be redundancies where consecutive frames are similar if the frames-per-second (fps) of the captured videos is high, which the authors do not specify. Train/test split should also be split video-wise, not frame-wise, which the reviewer found no information.


**Additional Feedback:**

None

**Clarity:**

The paper is structurally well-written and easy to follow.


**Correctness:**

The construction of the dataset is reasonable and considerate. The author tries to extend the variety of the dataset.

**Documentation:**

The authors provide sufficient documentation, including an URL. The authors provide sufficient configurations for reproducibility of the experiments.

**Ethics:**

The author claims to follow the FAIR Data principles and receive agreements from individuals in the video. All personally identifiable of non-volunteers are anonymized. Videos without the consensus of the people are discarded.

**Relation To Prior Work:**

The authors demonstrate a good comparison to other similar dataset, highlighting their strengths against others.

**Summary And Contributions:**

Authors release a large-scale dataset for fallen person detection in CCTV cameras. The dataset is curated under various conditions including lighting, camera heights and backgrounds. They perform an analysis on the metadata and feature-level distribution and demonstrate a good variability of the dataset. Experiments using popular object detection models including one-stage, two-stage and transformer-based were conducted to benchmark the performance in general and under different environment conditions.

---

> ### Author Response · Authors · 2021-09-27
> **Response to Reviewer 1eW1 (Part2)**
>
> We thank the reviewer for commenting on the ambiguities of video-based vs. frame-based data split. We also thought an unbiased and non-redundant split was an important issue. Therefore, we proceeded with a balanced video-wise split, where the total number of frames/videos with different characteristics and scenarios was roughly maintained in the same ratio to prevent bias. Indeed, we presented our balanced video-level statistics in Fig 4.  And, we will provide the video-wise split information in the final version for clarification.
>
> Again we thank the reviewer for the constructive comment, and we clarified this in Section 3.4.

---

> ### Author Response · Authors · 2021-09-27
> **Response to Reviewer 1eW1 (Part1)**
>
> **Q1.How would annotator categorize people during their falls (not completely on the ground)? There should be a clear definition for this binary classification problem.**
>
> We thank the reviewer for the constructive question. We use the following three rules to define the fall postures [1] followed by stumbling, bending down, limping, etc.:
>
> 1. Both shoulders should touch the ground (representing lie down, backward, and forward fall).
>
> or
>
> 2. One side shoulder touches the ground (representing slideway fall).
>
> or
>
> 3. When the upper body is occluded, the lower body from hips to foot touches the ground (sitting down from fall).
>
> **Reference**
>
> [1] Foroughi, Homa, Baharak Shakeri Aski, and Hamidreza Pourreza. "Intelligent video surveillance for monitoring fall detection of elderly in home environments." 2008 11th international conference on computer and information technology. IEEE, 2008.
>
> As long as one of the above three rules applies, we label it as a fallen event. Otherwise, we define it as a non-fallen event. Based on the above definition, we construct the binary classification problem. If there is ambiguity, five annotators manually examine and carefully cross-check to reach an agreement (at least more than half of the annotators need to agree). We add the above definitions in Section 3.2 for clarification.
>
>
>
> **Q2.Authors said the annotations for each frame are cross-check between 5 different students but did not specify the rule for interrater agreement and the criteria to accept an annotated frame.**
>
> Annotators cross-checked each other’s annotation to provide the consistency of labeling according to the following rules:
>
> 1. In the given frame, 1) the Occlusion labeling rule and 2) the Overlap labeling rule are strictly enforced.
> 2. A bounding box is correctly shown for the fallen person according to the fall event definition.
> 3. The label for the fall and non-fallen frames are correctly assigned.
> 4. Personally identifiable information (PII) is fully anonymized for non-volunteers.
>
> If there is a disagreement, all five annotators discussed and reached the agreement when at least half of them agreed.
> Again we thank the reviewer for the comment, and we added the details in Section 3.2.

---

### Official Review · Reviewer_UkCo · 2021-09-20
**VFP290K: A Large-Scale Benchmark Dataset for Vision-based Fallen Person Detection**

**Rating:** 7
**Confidence:** 3
**Correctness:** The technical aspects of the work see…
**Clarity:** The paper is very well written.

**Strengths:**

1. Presents a relatively diverse and extensive dataset on fallen person detection task. The dataset is publicly available for download and research purposes too.
2. The visualization and evaluations presented in the paper are very extensive.


**Weaknesses:**

1. One of the limitations of the work has to be a better motivation for the work. I understand that falls can be fatal and some of the numbers showcased in the introduction are well grounded on numbers from WHO and CDC. However, I would like some discussion as to how a fallen person detection task can come handy in avoiding some serious health related consequences in such situations.

2. While the most of the paper is very well written, I find the Ethics and Social Impact paragraph (in section 3.5) lacking some explanations. Can the authors put some lights on what sort of agreements were agreed upon by the volunteers and how has the anonymization process taken place for people who did not volunteer and were still in the frames (say while a volunteer is on the frame some non-volunteers might well be in frame too). Further clarification is also necessary on the line: ``we discarded the videos, if a person did not wish to be in.''.

**Additional Feedback:**

Please refer to weakness section.

**Documentation:**

Yes.

**Ethics:**

Please refer to point 2 in weakness.

**Relation To Prior Work:**

Yes, the same has been discussed clearly.

**Summary And Contributions:**

In this paper, authors create a novel, large-scale dataset (named VFP290K) for the detection of fallen persons composed of fallen person images collected in various real-world scenarios. They empirically demonstrate the effectiveness of the features through extensive experiments analyzing the performance shift based on object detection models.

---

> ### Author Response · Authors · 2021-09-27
> **Response to Reviewer UkCo (Part2)**
>
> We thank the reviewer for asking for clarifications. The details of consent and agreement forms are provided in Section 2 in Appendix. We fully informed the dataset collection procedures as well as a possible risk, discomforts, and withdrawal process to participants. The IRB at our institution thoroughly reviewed our consent form and data collection procedure, and the IRB acknowledged the minimal risk to our dataset collection process. We completely anonymized non-volunteers' personal identifiable information (PII) (i.e., faces, car number plates), using the anonymization method described in Section 5 in Appendix. In fact, we acknowledged our  mistake in the following sentence (**''we discarded the videos, if a person did not wish to be in.''**), and fixed the sentence as follows: We completely anonymized personally identifiable information (PII) of non-volunteers using the anonymization methods used in Section 5 in Appendix.
>
> We clarified and fixed this in Section 3.5. Once again, we appreciate the question from the reviewer to make our paper clearer.

---

> ### Author Response · Authors · 2021-09-27
> **Response to Reviewer UkCo (Part1)**
>
> We thank the reviewer for his/her careful comment. Detecting fallen people early can be critical, especially for the older people. The death rate of the elderly significantly increases if they are left untreated for a prolonged period of time [1][2]. Therefore, there is **a golden hour** to cope with, handle, and treat such falls that might cause severe injuries, and our system can be helpful for promptly responding to those.
>
> **Reference**
>
> [1] “Important Facts about Falls”, Centers for Disease Control and Prevention, National Center for Injury Prevention and Control 2017, https://www.cdc.gov/homeandrecreationalsafety/falls/adultfalls.html
>
> [2] Wild, Deidre, U. S. Nayak, and B. Isaacs. "How dangerous are falls in old people at home?." Br Med J (Clin Res Ed) 282.6260 (1981): 266-268.
>
> Hence, the fallen person detector trained with our dataset can provide rapid response to notify medical staff or caregivers to properly handle emergencies and prevent further serious injuries from falls. Our dataset can be used as a part of intelligent healthcare monitoring systems integrated with a smart CCTV [3] that autonomously recognizes the fall events. Especially, in the presence of increasing lonely deaths [4,5], our approach can be beneficial for the elderly who are left at home alone [6], as well as public health monitoring in open spaces [7].
>
> Additionally, our dataset can be used for fallen person detection from violence and crimes, where the prompt responses to the falls caused by violent events can also be detected. Our envisioned integrated monitoring system can also notify law enforcement and emergency responders to prevent further injuries, which require rapid responses.
>
> We clarified and elaborated above points in Section1, as requested by the reviewer.
>
> **Reference**
>
> [3] Mohan, H. M., et al. "Edge Artificial Intelligence: Real-Time Noninvasive Technique for Vital Signs of Myocardial Infarction Recognition Using Jetson Nano." Advances in Human-Computer Interaction 2021 (2021).
>
> [4] Coronavirus Victims Are Dying Alone, WSJ, 2021 https://www.wsj.com/articles/coronavirus-victims-are-dying-alone-11586088001
>
> [5] Even in a Pandemic, No One Should Have to Die Alone, WSJ, 2020, https://www.wsj.com/articles/even-in-a-pandemic-no-one-should-have-to-die-alone-11586557751
>
> [6] Foroughi, Homa, Baharak Shakeri Aski, and Hamidreza Pourreza. "Intelligent video surveillance for monitoring fall detection of elderly in home environments." 2008 11th international conference on computer and information technology. IEEE, 2008.
>
> [7] Wang, Wenjin, et al. "Guest Editorial: Camera-Based Monitoring for Pervasive Healthcare Informatics." IEEE Journal of Biomedical and Health Informatics 25.5 (2021): 1358-1360

---

### Author Response · Authors · 2021-09-27
**Paper Revision: Summary of updates and thanks.**

We thank all the reviewers for their insightful comments and constructive questions. We have thoroughly reviewed all the comments and updated the manuscript, and highlighted the changes in yellow. And the summary of the changes are provided below:

- We have added additional details on the motivation of this work in Section 1 for fallen person detection.

- We have provided more details and clarifications on the agreement and anonymization process during the data collection and preprocessing step.

- We have clarified the detailed annotation, quality control, and cross-checking process by explaining the precise definition of the fall cases and the annotation rules.

- We have further clarified the video-wise vs. frame-based split and training/test ratio in Section 3.

- We have discussed the limitations and challenges and further elaborated on the bias and diversity issues.

Again, we are very grateful for the reviewers’ comments and suggestions, finally allowing us to revise the paper. We believe our responses and the revised manuscript can clarify the questions and comments from reviewers. Please let us know further feedback.

---

### Decision · Program_Chairs · 2021-10-09

**Decision:**

Accept

**Comment:**

The reviewers acknowledge the novelty of the fallen person detection dataset. Although it has some drawbacks such as limited number of subjects and acted events, this is very difficult to have real data for the problem. Overall, the data and provided analyses show unique aspects and this can be a good resource for the community.